

# Artificial night light alters nocturnal prey interception outcomes for morphologically variable spiders

Suet Wai Yuen and Timothy C. Bonebrake

School of Biological Sciences, The University of Hong Kong, Hong Kong

## ABSTRACT

Artificial night light has the potential to significantly alter visually-dependent species interactions. However, examples of disruptions of species interactions through changes in light remain rare and how artificial night light may alter predator–prey relationships are particularly understudied. In this study, we examined whether artificial night light could impact prey attraction and interception in *Nephila pilipes* orb weaver spiders, conspicuous predators who make use of yellow color patterns to mimic floral resources and attract prey to their webs. We measured moth prey attraction and interception responses to treatments where we experimentally manipulated the color/contrast of spider individuals in the field (removed yellow markings) and also set up light manipulations. We found that lit webs had lower rates of moth interception than unlit webs. Spider color, however, had no clear impact on moth interception or attraction rates in lit nor unlit webs. The results show that night light can reduce prey interception for spiders. Additionally, this study highlights how environmental and morphological variation can complicate simple predictions of ecological light pollution's disruption of species interactions.

## INTRODUCTION

Artificial night light represents an emerging threat to biodiversity (*Hölker et al., 2010*). Studies are increasingly demonstrating the widespread impacts of artificial light on species (*Longcore & Rich, 2004*), ecological communities (*Davies, Bennie & Gaston, 2012*; *Gaston et al., 2014*) and ecosystem functioning (*Macgregor et al., 2017*). One important consequence of light pollution and artificial night light is a change in natural light landscapes and seascapes that could alter species communication and interactions (*Davies et al., 2013*). Within species, visual communication can be disrupted by ecological light pollution with consequences for mating success and population dynamics (*Firebaugh & Haynes, 2016*). However, the prospect that artificial light could disrupt visual cues, color perception and species interactions remains an important subject for further empirical study (*Delhey & Peters, 2017*).

Corresponding author
Timothy C. Bonebrake,
tbone@hku.hk

Predator–prey interactions in particular are potentially vulnerable to ecological light pollution. Prey species often rely on specific light conditions for camouflage or crypsis where artificial night light can render such species detectable by predators (*Davies et al., 2014*; *Underwood, Davies & Queirós, 2017*). Light pollution can also impact prey behavior including vigilance effectiveness (*Yorzinski et al., 2015*) and predator avoidance (*Perkin et al., 2011*). Similarly, *Minnaar et al. (2015)* found that bats in lit areas consumed six times more moths than bats in unlit areas, highlighting the importance of night light in attracting moths and dramatically changing the dynamics of a long-evolved predator–prey relationship. Some spiders also appear to take advantage of light attraction of prey species and preferentially choose web building sites near light sources (*Heiling, 1999*), potentially explaining why some orb-weaver spiders have successfully established within well-light urban areas (*Lowe, Wilder & Hochuli, 2014*).

Coloration in spiders is particularly important and recent studies have shown that nocturnal spiders can use body markings as visual lures to attract prey (*Tso et al., 2006*; *Chuang, Yang & Tso, 2008*; *Fan, Yang & Tso, 2009*; *Zhang et al., 2015*). Night vision is a challenge for nocturnal species requiring visual systems that exploit moonlight, zodiacal light, airglow and starlight to distinguish color and detect motion (*Warrant, 2004*; *Cronin et al., 2014*). Therefore, nocturnal insects lured to spider coloration might be differentially attracted or deterred under artificial night light.

To examine the potential of artificial night light in disrupting a visually driven predator–prey relationship, we manipulated light conditions and spider body coloration in a field experiment of the orb weaver spider, *Nephila pilipes*. The spider has a distinctive yellow body pattern that is well known as a visual lure for nocturnal prey species (*Tso, Lin & Yang, 2004*; *Fan, Yang & Tso, 2009*). We predicted that artificial light would decrease the effectiveness of the visual lure and impact prey attraction outcomes for spiders.

## MATERIALS AND METHODS

### Study sites and species

*Nephila pilipes* are orb-weaver spiders commonly found in Hong Kong during the wet season (March to November). The species exhibits conspicuous yellow patterns scattered over their dorsal and ventral cephalothorax, abdomen and legs. *N. pilipes* hunt diurnally and nocturnally and use these conspicuous color patterns to attract prey (*Tso, Lin & Yang, 2004*). Habitats include semi-rural areas in Hong Kong where artificial night light sources from street lamps and housing are common. Six sites were chosen due to their semi-rural characteristics including outlying islands (Lamma Island and Lantau Island), country park forests (Tai Tam, Parker Hill and Pak Tam Chung) and a rural village (So Kwun Wat) (Fig. 1). Sites were close to low-density residential areas where active human disturbance was low relative to nearby urban areas.

### Experimental design

In each site, we searched for adult female spiders in their webs (females are much larger than males in *N. pilipes* and display the visual lures of interest). We chose four spider individuals at each site and each adult was assigned randomly to one of four experimental
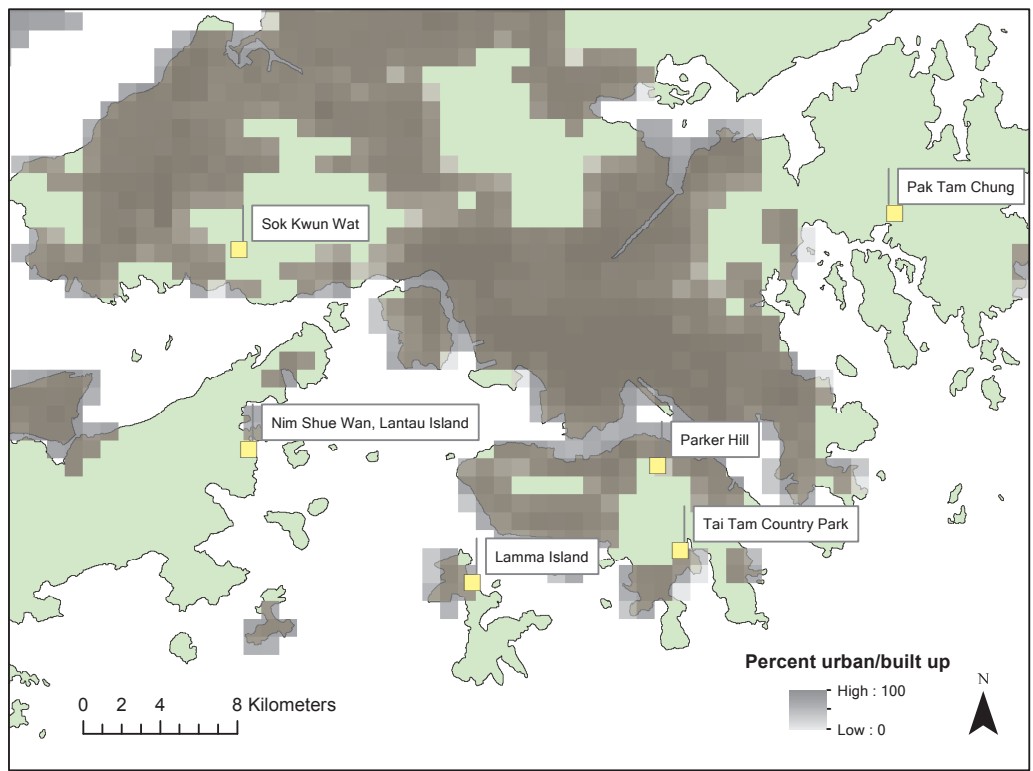

**Figure 1** **Map of the study sites across Hong Kong.** Percent urban area is based on the one km resolution land cover dataset of *Tuanmu & Jetz (2014)*.

setups: (1) spider web was lit by a lamp and yellow spider coloration was not painted away (lit control), (2) yellow spider coloration was painted away by black paint (dark and painted), (3) both light and paint were applied to the spider (lit and painted), and (4) no lamp was present and yellow coloration remained (dark control). All setups within a site were located within 50 m of each other. We used magnetic USB mini LED bulbs for the light source (WERTIOO) directed towards the web (approximately one meter from the web) as a light source in lit setups, with paper tape wrapped over the surface to downward adjust the light intensity. These bulbs have a similar emission spectrum as Hong Kong street lamps (i.e., high pressure sodium lamps), characterized by peaks in wavelength between 450 and 600 nm (*Elvidge & Keith, 2009*). For spider color manipulations, we used black acrylic paint due to its low toxicity and ease of removal (Tamiya Colour Acrylic Paint Mini Black X-1, diluted by thinner of the same brand). We also applied the paint to black body parts of spiders in the yellow coloration "controls" (setups 1 and 4) to exclude possible effects of ink. Spiders were gently removed from their webs and cooled to an inactive state by an ice bath. After treating with paint, they were carefully placed back on their webs for the commencement of the experiment. Spiders generally regained motility about one minute after withdrawal from the ice bath and returned to an un-agitated state several minutes after that.

We also recorded body length, the total length of cephalothorax and opisthosoma, of each spider. Web size was measured as the average of web radii in eight cardinal directions, from web hub to the outermost capture spiral following *Tso et al. (2006)*. We measured light intensity once for each treatment by placing a digital light meter (Dr. Meter LX-1330B; Hisgadgets, Union City, CA, USA) one meter (as close to perpendicular to the web as possible given environmental constraints) from the webs at the beginning of the experiment.

We examined *N. pilipes* predation success through single-night videotaping of all experimental spider webs. We recorded from between 7:00 p.m. and 8:00 p.m. to the beginning of sunrise (4:00 a.m. to 5:00 a.m.). We sampled each site once in the fall (August to November) of 2015 and 2016. All setups in each site were recorded on the same night simultaneously. Environmental conditions (moon phase, cloudiness, temperature, etc.) are certain to vary over time and could affect prey attraction and interception rates—however, because all treatments per site were done on a single night we largely controlled for these factors (these factors were unlikely to vary much *within* sites). As a trade-off for controlling conditions within sites, the proximity of each treatment within each site allowed for the possibility of moths in one treatment to be affected by another treatment (e.g., lit web attracting moths from an unlit web). We used high definition car rearview cameras with infra-red emitters (Theera YRS0889) for the night video recording. *N. pilipes* mostly builds webs under tree trunks, at a height of one to two meters. Cameras were therefore placed on tripods approximately one meter from each web.

We examined prey responses to each treatment by focusing on moths in all video recordings. Moths are a dominant nocturnal prey species of *N. pilipes* (*Fan, Yang & Tso, 2009*). Moths also use a trichromatic visual system, possessing green, blue and ultra-violet light receptors (*Briscoe & Chittka, 2001*). Based on a model of a hawkmoth visual system (*Johnsen et al., 2006*), *Chuang, Yang & Tso (2007)* determined that the yellow stripes of *N. pilipes* were more distinctive under moonlight than black markings. Specifically, the contrast of black body parts of the spiders with vegetation backgrounds was remarkably smaller than those of yellow body parts (*Chuang, Yang & Tso, 2007*). Yellow flowers could represent food sources for some moth species such that this coloration pattern might be attractive (*Johnsen et al., 2006*).

## Data analysis

We focused on three variables for analysis; attraction (number of moths seen approaching the web), interception (number of moths that physically hit the web, but not necessarily caught in the web) and interception efficiency (number of moths intercepted divided by the number of moths attracted). We used generalized linear mixed models to model the moth attraction and interception data. We fit the models with a negative binomial distribution due to the nature of the count data and the fact that the data were over-dispersed (*Zuur et al., 2009*). We included site as a random effect and number of hours as an offset term. We modeled web size, color (categorical: painted or not painted), light (categorical: lit or unlit) and spider size as fixed effects and modeled all combinations and first order interactions. We log transformed interception efficiency and used a linear mixed effects model with site

**Table 1 Model results for attraction, interception and interception efficiency.** We used generalized linear mixed models (GLMM) for attraction and interception and linear mixed effects models (LMM) for interception efficiency. Fixed effects include color, light, spider size (Size), and web size (Web).

| Models | Log-likelihood | $AIC_c$ | $AIC_c$ Delta | Akaike Weight |
|---|---|---|---|---|
| **Moth attraction (GLMM)** | | | | |
| Color, Light, Size, Color:Size | −145.14 | 312.9 | 0.00 | 0.61 |
| **Moth interception (GLMM)** | | | | |
| Light | −95.47 | 201.4 | 0.00 | 0.31 |
| Light, Size | −94.29 | 202.6 | 1.13 | 0.18 |
| Color, Light, Size, Color:Size | −90.65 | 203.9 | 2.48 | 0.09 |
| Light, Web | −95.18 | 204.4 | 2.92 | 0.07 |
| Color, Light | −95.39 | 204.8 | 3.35 | 0.06 |
| Light, Size, Web | −93.64 | 205.3 | 3.84 | 0.05 |
| Color, Light, Size | −93.665 | 205.3 | 3.89 | 0.04 |
| **Interception efficiency (LMM)** | | | | |
| Light | −20.83 | 52.2 | 0.00 | 0.35 |
| Size | −21.56 | 53.6 | 1.47 | 0.17 |
| Light, Size | −19.90 | 53.8 | 1.65 | 0.15 |
| Color | −22.17 | 54.8 | 2.68 | 0.09 |
| Color, Size, Color:Size | −18.70 | 55.4 | 3.24 | 0.07 |

as a random effect and web size, spider size, color and light as fixed effects. For the linear mixed effects models, we also examined all first order interactions. All models were run in R using packages glmmADMB and nlme (*Pinheiro et al., 2014*; *Skaug et al., 2014*). For each set of generalized linear mixed models and linear mixed effects models, we used model selection and chose the model with the best AICc. For all models where delta AICc <4, we used a model averaging to determine relative variable importance (RVI) using package MuMIn (*Barton, 2013*).

## RESULTS

We recorded 163 h during the course of the study period. In three cases (two unlit painted treatments and one lit unpainted treatment) the spider left and so we were unable to collect data. Over the entire recording period we documented 5,375 moths attracted and 300 moths intercepted—but only ten moths remained in the web and were consumed by the spider (nine of the ten moths consumed by spiders were done so by unpainted spiders).

The best model for attraction included light, color, spider size and the interaction between spider size and color (Table 1; light: estimate ± se: −2.40 ± 0.51, $P < 0.001$; color: −13.1 ± 2.91, $P < 0.001$; size: −0.18 ± 0.45, $P = 0.69$; size:color: 2.65 ± 0.63, $P < 0.001$). In sites with smaller spiders, fewer moths flew near the webs (Fig. 2: for large spiders above 4.5 cm, 31.9 ± 6.1 (mean ± standard error) moths attracted/hour and 14.1 ± 4.7 for small spiders).

For interception, lit webs had lower interception rates (Fig. 3) and the best performing model included only light as a variable (Table 1; light: estimate ± se: −2.47 ± 0.88, $P = 0.005$). Seven models performed equally well (Table 1), and of those models light
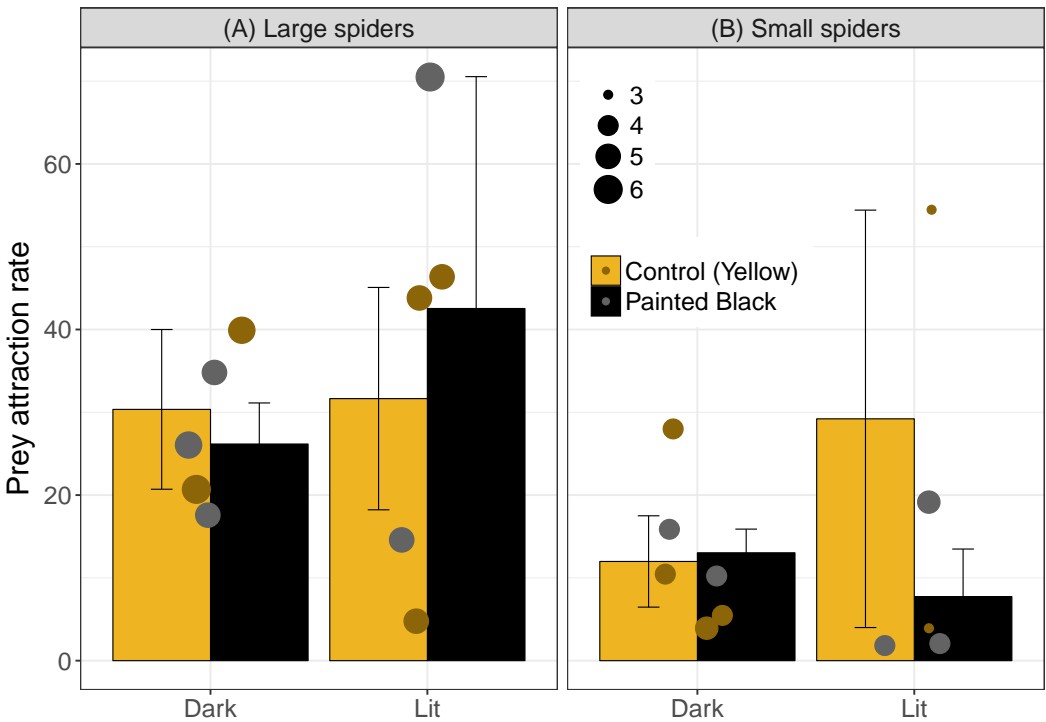

**Figure 2** **Prey attraction rates (number of moths/ hour) across color and light treatments.** Error bars indicate mean ± standard error. Spider size was broken down into categorical variable for visualization, where spiders above 4.5 cm were identified as large (A) and were otherwise identified as small (B). Data points are scaled to spider size (in cm).

(RVI = 1.0) was the most important followed by spider size (RVI = 0.45) and color (RVI = 0.24).

For interception efficiency, the best three models were light alone, size alone and light plus size (Table 1; light alone: estimate ± se: −0.43 ± 0.25, $P = 0.10$; size alone: −0.33 ± 0.18, $P = 0.08$; and light+size: light: −0.54 ± 0.26, $P = 0.05$; size: −0.37 ± 0.17, $P = 0.04$). Of the five best performing models (Table 1) light was the most important variable (RVI = 0.61) followed by spider size (RVI = 0.47) and color (RVI = 0.19). Light tended to diminish interception efficiency (especially for webs with large spiders) and small spiders exhibited a higher efficiency (Fig. 4). For large spiders, those that were unpainted (yellow) and in dark webs had a higher interception efficiency than all other treatment combinations (Fig. 4).

## DISCUSSION

Prey attraction rates were affected by spider size, spider color manipulation, and light conditions but not in a consistent manner (Fig. 2). We did however find a consistent and large effect of light in lowering prey interception (∼1 moth/hr; Fig. 3) comparable to the positive effect of yellow markings documented in previous studies of *N. pilipes* (∼1 moth/hr; *Chuang, Yang & Tso (2007)*). In this study, we found no clear consequences

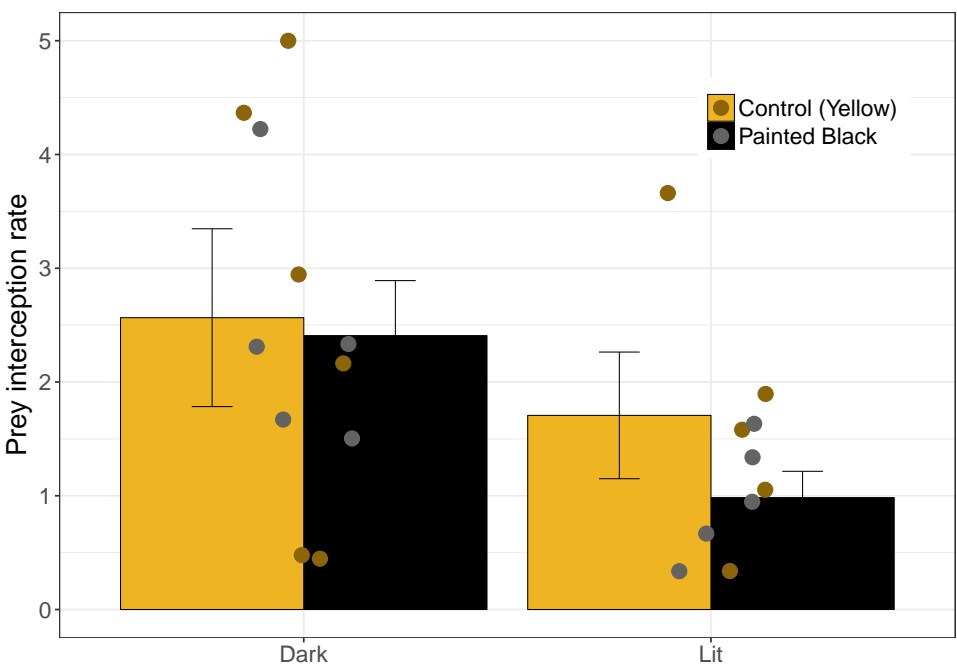

**Figure 3** **Moth interception rate as a function of spider color and light.** Error bars indicate mean ± standard error.

of spider color for prey attraction or interception. Yet, spider size and color exhibited an interactive effect for all attraction variables (Table 1) demonstrating the complexity of the relationship between spider morphology and prey attraction outcomes. The lack of clear effects of some variables (e.g., color) on attraction and interception might be a consequence of the relatively low sample sizes of the experiment (six experimental units/ nights). Nevertheless, our results highlight (1) that light can reduce prey interception for spiders and (2) the role of morphological and environmental variation in complicating and potentially obscuring important but difficult-to-detect artificial night light effects on predator–prey interactions.

We found that interception rates were lower in lit webs than unlit webs. The lack of an effect of color on interception suggests that the cause for this pattern is unlikely through changes in the effectiveness of the visual lure. Moth flight-to-light responses can be varied and likely determine light's effect on interception rates. Chaotic and undirected flight towards light sources in moths (*Frank, 1988*) may reduce the effectiveness of webs in prey capture for lit webs. The presence of light will also change the contrast of the web with its background and could make the web more easily perceived and avoided by moth prey (*Craig, 1988*; *Théry & Casas, 2009*). Alternatively, while some spiders build webs near artificial light sources to take advantage of attracted prey (*Heiling, 1999*), there might be a tradeoff in web detectability that could offset increased prey availability near light. Further exploration into the costs and benefits of locating webs near artificial night light

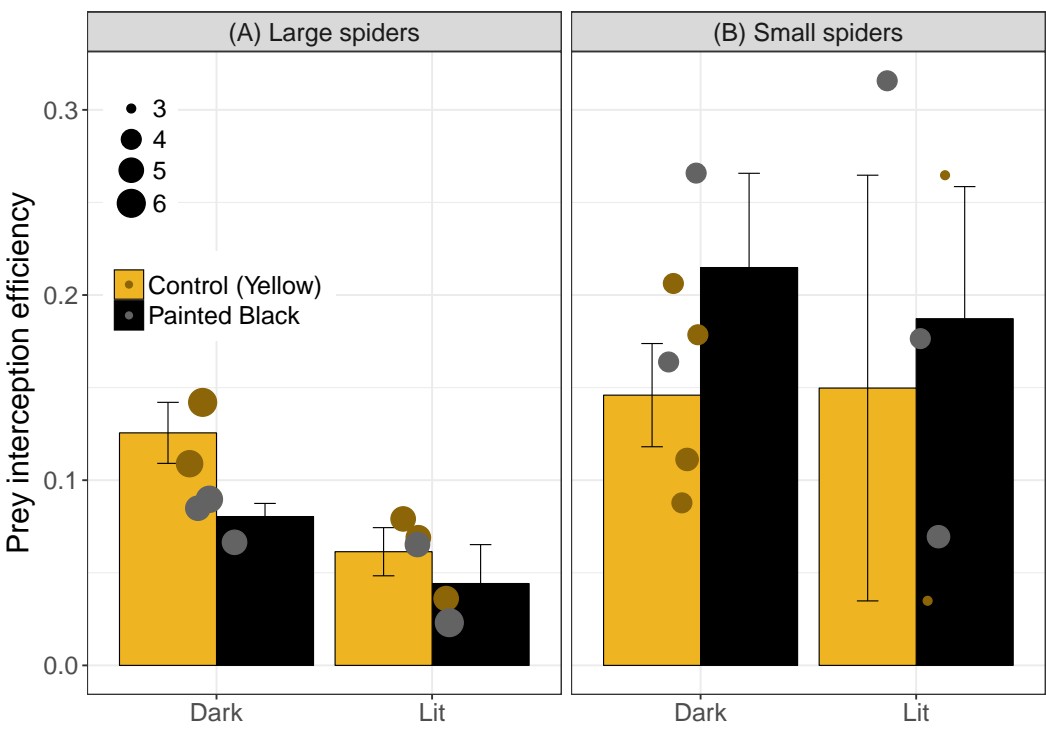

**Figure 4   Interception efficiency across color and light treatments.** Error bars indicate mean ± standard error. Spider size was broken down into categorical variable for visualization, where spiders above 4.5 cm were identified as large (A) and were otherwise identified as small (B). Data points are scaled to spider size (in cm).

would be productive in revealing the impacts of ecological light pollution on urban spider populations.

For attraction, the light treatment alone did not consistently result in higher attraction rates. While each site had all treatments on the same day to minimize time or site effects on the results, the high variation in attraction may have obscured possible effects of light and morphology. We also only used one type of bulb and higher light intensity sources may have led to clearer results. Light can have strongly contrasting effects on predator–prey interactions depending on the habitat (*Russ, Lučeničová & Klenke, 2017*). Environmental conditions across sites and days, and variable light intensity in particular (e.g., different moon phases), may have resulted in complex patterns between the light treatment effects and prey attraction. We did find that, in general, larger spiders had more moths attracted to their webs than smaller spiders (Fig. 2). The higher attraction rate for large spiders could be a consequence of a bigger visual lure capable of attracting more prey than smaller spiders (*Hauber, 2002*). Alternatively, spiders who select better microhabitats with more prey may grow larger (*Brown, 1981*) such that the association is driven by greater prey availability increasing spider size, not larger spiders attracting more prey. In any case, the high heterogeneity of light landscapes across spatial scales in the environment (*Swaddle et al., 2015*) along with other sources of microhabitat and temporal variation are important
considerations for the results of this study as well as ecological light pollution studies broadly.

Size and color likely have different effects at variable distances: small spiders may attract fewer prey (or be located in sites with fewer prey) but they also tend to have a higher (but variable) interception efficiency suggesting that their small size might be advantageous in disguising their presence once moths are near the web. For large spiders, the unlit and unpainted (yellow markings intact) treatments had the highest interception efficiency which could be indicative of increased, effectiveness of the lure for large spiders in natural light conditions. However, interception efficiencies across treatments were highly variable across treatments and the mechanisms and effects are difficult to interpret. In addition to the importance of morphology for prey attraction, the visual cues and coloration of *N. pilipes* may also attract their own predators potentially (*Fan, Yang & Tso, 2009*; *Meyer & Sullivan, 2013*; *Yeh et al., 2015*). Artificial night light effects on predator–prey relationships must then take into account multiple morphological variables (e.g., color and size) as well as multiple consequences of those visual cues (e.g., prey attraction and predator attraction).

Temporal consequences of artificial night light are worth consideration in this case. For *N. pilipes*, *Chuang, Yang & Tso (2007)* found that prey interception was generally higher at night but that diurnal prey interception occurred as well. Light pollution may then shift hunting activity of *N. pilipes* and increase the importance of diurnal prey interception. Circadian responses to artificial night light, particularly in the context of predator–prey relationships, remain an important avenue of future research (*Dominoni, Borniger & Nelson, 2016*).

Our results provide insights into the relationship between prey attraction and artificial night light but this study may not be reflective of the entirety of night light impacts on this predator–prey relationship. For example, the short time period covered for each web in this study could miss crucial rare catches of large prey items which may have disproportionate positive impacts on spider fitness (*Venner & Casas, 2005*). If light differentially affects or attracts large moths (*Wölfling et al., 2016*) then the average overall interception rate may be less important than the ability to catch large moths occasionally. Similarly, more important than interception rate, is the rate at which spiders actually consume prey. Though we only observed ten moth consumption events, interestingly, nine of those moths were captured by spiders that were not painted. The effects of artificial night light on predation rates are rarely straightforward and will ultimately be the product of both negative and positive impacts of light on predator and prey behavior (*Grenis, Tjossem & Murphy, 2015*).

Reduced feeding, mobility and possible increased predation are all potential consequences of artificial night light for moths and these effects may result in trophic cascades within these novel ecosystems (*Van Langevelde et al., 2017*). This study highlights how night light can impact species interactions but also that environmental and morphological variation can obscure simple relationships between predator and prey. As artificial night light continues to dramatically alter the environment (*Davies et al., 2013*), the conditions under which these predator–prey relationships have evolved will also change resulting in the possible disruption of important species interactions within ecosystems.

## CONCLUSION

Our experiment demonstrates a clear reduction of prey interception in webs with artificial night light for orb-weaver spiders in Hong Kong. The results also suggest that body size and color might also influence prey interception outcomes for orb-weaver spiders but no clear patterns were detected. Future research into the complex interactions between predator, prey, light and morphology will aid in predictions and understanding of how anthropogenic changes in light are likely to affect ecological communities and ecosystems.

## ACKNOWLEDGEMENTS

We thank Schind Lee and Danny Yuen for their assistance in the field and for their help in managing unpredictable field situations. Jon Bennie and two anonymous reviewers provided useful input that improved the manuscript significantly.

### Funding
The authors received no funding for this work.

### Competing Interests
The authors declare there are no competing interests.

### Author Contributions
- Suet Wai Yuen conceived and designed the experiments, performed the experiments, analyzed the data, wrote the paper, prepared figures and/or tables, reviewed drafts of the paper.
- Timothy C. Bonebrake conceived and designed the experiments, analyzed the data, contributed reagents/materials/analysis tools, wrote the paper, prepared figures and/or tables, reviewed drafts of the paper.

### Field Study Permissions
The following information was supplied relating to field study approvals (i.e., approving body and any reference numbers):

Most study sites did not require any permit for access. For the two country parks (protected areas) a permit issued to the School of Biological Sciences (HKU) by HK Government (AFCD) covered the scope of work within the country parks.

### Supplemental Information
Supplemental information for this article can be found online at http://dx.doi.org/10.7717/peerj.4070#supplemental-information.

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
