# Peer review of "Artificial night light alters nocturnal prey interception outcomes for morphologically variable spiders"

_PeerJ, doi:10.7717/peerj.4070_

## Round 0.1 · original submission · Minor Revisions

Three experts in your field have reviewed your manuscript and all provided generally positive reviews of your study. All three reviewers request greater clarity and detail in your description of your Methods. Reviewer 1 also suggests that you provide more information about the natural history and spectral sensitivity of your study species, which I encourage you to do. Additionally, Reviewer 2 raises some concerns about the treatment of your data and your analyses that you should address. I particularly agree with Reviewer 2’s comment about your choice to place the spiders into two categories based on size. Please describe this decision in your methods/analysis section and explain your reasoning as well as how the two size categories related to the average size of your test population.


In addition to the reviewers’ comments and questions, I have a few of my own.

I would like to know more about the six sites were you sampled the spiders as this may in turn have impacted the local density of moths and/or the behavior and site choice of the spiders. Can you provide information about how independent the sites were. Specifically, how far apart were the six sites from one another? Do you have information about the degree of urbanization at each site (e.g., % pervious ground cover, or density of housing or other anthropogenic elements such as street lighting)?

I think that a photograph or diagram of your experimental set up would be useful. Or perhaps provide exemplar footage from your moth inception videos as supplemental information.

How quickly did the spiders regain motility after the ice bath? Would such a treatment have impacted their typical behavior?

What was the moon phase for each of your test nights? Was it consistent across testing?

You state “In three cases the spider left and so we were unable to collect data” which of your four experimental conditions did these represent?

Finally, I have some more minor editorial suggestions:

Line 169, for clarity, please reword thus: “Prey attraction rates were affected by spider size, spider color manipulation, and light conditions…”

Line 173, for clarity, please reword thus: “we found no clear consequences of spider color for prey attraction or interception”

Line 175, for clarity, please reword thus: “the complexity of the relationship between spider morphology and prey attraction outcomes”

Line 206, the sentence that starts “For large spiders, that the unlit and unpainted (yellow markings intact) treatment…” is somewhat awkwardly phrased. I suggest rephrasing for clarity.

Table 1 caption has a typo at the end “Roman"; coᤲ �BG” Please remove.

Reviewer 1 ·

Basic reporting

I enjoyed reading the article of Yuen and Bonebrake. I found it well-written, clear and sound. The authors tested the possibility that artificial light at night could disrupt the hunting success of weaver spiders, which usually hunt by luring flying insects into their webs by means of colour attraction, mimicking with their yellow colour the appearance of flowers. They predicted reduced hunting success under artificial light at night. I found the interpretation of results sound, well-founded and well-referenced. I have however several remarks about the experimental set-up and the description of the methodologies used, which should be addressed clearly for this article to be acceptable for publication.

Experimental design

Generally, the description of the light measurements is very poor. In particular:

1) When were the measurements done? Which days of the experiment, and at what time?
2) Was light intensity measured only once per site, or repeatedly? This is important to understand as repeated measurements can reduce the influence of other environmental variables on the results (cloud cover, temperature, etc).
3) The authors mentioned the light type used, but do they know anything about the actual wavelength? Was this ever measured?
4) Similarly, what was the exact light intensity measured at each experimental web?
5) It is unclear what the variable "light" exactly represent in your models. Is this the actual light intensity or is it a categorical variable (light-control)? If the latter, why not incuding the actual light intensity as a predictor?

I further think that you should recognise and emphasize in the discussion that the results, although significant in some cases, are based on are relatively low sample size.

Also, it would be good to know something more about the species you used in terms of:

1. Is there anything know about the spectral sensitivity of this spider species?
2. You mention (L81) that this species hunt both diurnally and nocturnally. Is there anything know about its actual circadian system? It sounds as if it possesses a very flexible circadian rhythm, which might actually be beneficial in the context of light pollution if they could switch to a more diurnal hunting style. The important of knowing more about circadian rhythms in such a species, and its sensitivity to light, might be stressed more, with some relevant literature added (Gaston et al 2013 Biological Reviews, Dominoni et al 2016 Biol Lett, Davies et al 2013 Global Change Biology).

Validity of the findings

As stressed in the previous box, more details on the methodologies and background on the species in terms of light sensitivity and circadian rhythms could be beneficial to the overall quality of the paper.

Reviewer 2 ·

Basic reporting

The study is very interesting and addresses questions on a hot topic in ecology. I enjoyed how the background knowledge is presented in the introduction and how the authors justify the relevance of the questions they address. Also literature is properly cited and relevant. Description of the figures is good, as well as the labelling, nevertheless I would improve them, but see my comments on the following sections of the review. The manuscripts is written in proper English and well comprehensible and the data are supplied.
This said, I have some major issues that, I think, justify my recommendation for the editor.

Experimental design

I find the study novel, relevant and interesting and I like the design which is also well described. The investigation has rigorous principles but I see important issues, both conceptual and methodological, that undermine the result interpretation and the main conclusion of the study.
Firstly I want to make sure that the authors considered few points that are not mentioned in the manuscripts but are important for the relevance of the study.
1. From the data supplied I see that light intensities in the illuminated orb-webs present are highly variable. If the illumination equipment has been installed in the same manner among webs measurements should be comparable. Of course spiders can build webs in very different fashions making a standardized installation difficult, but I would have tried to standardize the illumination since the sample size is not high enough to make inference on different light conditions. Also, the light intensity detected by the sensor of the measurement device can be very different according to the angle of incidence of the light beam and the distance from the light source, potentially leading to critical bias. Did you control for it? Please provide more explanations on these points.
2. Can you consider the six sites comparable? In the manuscript it is not mentioned whether specific features of the sites are taken into account. For example moon, sky glow, light pollution coming from close by roads, cities, micro conditions around the web (bushes, branches) that could shade those light sources. Being moths highly mobile animals, overall lighting of the site might be relevant. It would be good if these conditions are standardized among sites in order to assume environmental homogeneity.
3. Overall moth abundance and activity and phenology would greatly affect your results. In my experience moth activity can dramatically vary among and within seasons depending on winter and early spring meteorological conditions (larvae development and survival), blooming of some key plant species (within season phenology), temperature, humidity and moon (within night activity). You collected data during one season, each site was sampled four (or three) times simultaneously within one night; this lead me to wonder how did you control for moth abundance and activity in your results.
4. As already mentioned, moths are highly mobile animals and can cover several kilometres in a single night. I therefore ask the authors to justify the independence of the spiders you selected within a site. Albeit I believe that the light treatments do not affect the light conditions of the dark treatments, I am afraid that moth are. For example, you could detect less attracted or intercepted moths in a dark treatment because moths are attracted to the close by light treatment. On the other hand you could detect more moths on a dark treatment because moths are rejected by the close by light treatment. This is probably species dependent and, in order to make proper inference, you need to ensure independence.

Validity of the findings

The validity of the findings are undermined by two types of issues: statistical and biological. Concerning the statistical part:
1. the sample size is certainly low and there are no replicates within sites. To make proper inference a higher N and replicates are desirable, for example by sampling the same spiders several nights, looking for more spiders within the same site, increasing the site number and repeating the data collection over two or more years. This point is linked to point 3. in the previous section of this review where I stated the need to control for moth activity among and within season. It is often very difficult to achieve a satisfying sample size in a field study, and if sites can be considered as homogeneous I think that it should not be an insurmountable problem. However, this is a weakness of the study and might seriously undermine the robustness of your results.
2. Spider size is modelled as a quantitative variable but visualized (in figures 1 and 3) and discussed as a categorical. This makes things even harder to understand and interpret. Also, the body size threshold that splits between small and large spiders is arbitrary and this should be stated in the text, or justified otherwise. I am wondering how the results change if the authors model the variable spider size as a categorical with two levels (small and large).
3. When dealing with quantitative variables (size, light intensities, attraction and interception rates, interception efficiency) it is generally desirable to represent data with scatter plots. Consider it in your next version of the manuscript.

Beyond the lack of satisfying sample size and of some important predictors (see previous section of this review), there are important conceptual issues that I would like to highlight.
1. I am puzzled by the response variables the authors are considering, especially by the interception rate. What is the biological relevance of it? If only a tiny part of the intercepted moths are eventually captured and consumed, what is the effect of light and painting of the lure on capture rate, moth mortality and spider food intake? In other words, how can attraction and interception affect prey-predator interactions? How could this set up and the results provide clues about the actual pressure of an anthropogenic disturbance such as artificial night light on the ecological interaction the authors are considering? Please, could you specify in which treatments and sites the ten predatory events occurred?
2. Moths flying close to lit webs could avoid them because they can better detect them, resulting in reduced interception events. However, their flying pattern could be greatly disrupted by artificial night light inducing them to aimlessly fly around the lamp avoiding the spider web. On a normal, aimed movement moths might be more likely to hit a spider web. Please, mention the well known fly-to-light behaviour when you discuss your results.
3. What is the time span of the sampling? There must be a typo in line 112. If the fieldwork has been conducted during one month (between Oct 2016 and Nov 2016) I do not see additional problems. If on the other hand several months elapsed from the first to the last sampling, a time variable musts be modelled.

Additional comments

The story around this study is beyond any doubts fascinating, the topic is hot and the design cleverly conceived. However, as the authors fairly mention in the discussion, the outcome of the study is difficult to interpret and does not provide conclusive results. It gives clues about the role of the lure on the chephalothorax of Nephila pilipes and the effect of artificial night light on the phases prior to the predatory event. Unfortunately, it does not highlight changes in moths' capture rate and spider food intake making the study of poor biological relevance. Further, the low sample size leads to a poor statistical power, makes the model selection difficult, results unclear and the conclusions unsatisfying.
However, I believe that the value of the study can be enhanced. I propose that the authors provide a power test to ensure that the data support the results and that the manuscript will be rewritten in order to better explain the data. Important points that need to be discussed are the definition of prey-predator interaction and the biological relevance for the animals involved in the interaction. If available, additional data could be added: environmental measurements like temperature and humidity, characterization of the light conditions in each site. An important control would be to sample data on lit and unlit orb-web where Nephila pilipes is absent to further disentangle the role of the lure which I believe is important. Light intensities in lit treatments are relatively low if you consider what you would detect within few meters from a street lamp. Increasing light intensity you could get stronger effect of the light.

Additional comments and questions on the manuscript:
Line 90: why only females?
Line 112: typo in the starting and ending date?
Line 125: are yellow flowers a common moth food source? According to my knowledge, flowers specialized on nocturnal pollinators do not invest much on striking pigmentation and are therefore mostly whitish/pinkish. However, I am not a specialist of your study area.
Line 236: limit the conclusion to the study species or justify the extension of the conclusions to other orb-weaver spider species that might be devoid of lure.

·

Basic reporting

The paper reports clearly on the methods and results of an interesting experiment. It is well-written, with good reference to the literature and well structured.

Experimental design

The experimental design is good.
I have two points that should be added to the methods section if possible - first, at the least the manufacturer and the colour temperature (in K) or ideally, the spectral power distribution of the LED lamps used should be included in order to allow the experiment to be replicated. If possible, the sectrum of the lights should be compared to the high pressure sodium lights used in Hong Kong that the authors are trying to replicate.
Second, the time of sunset during the study should be stated.

Validity of the findings

The data is robust and analysis statistically sound, and the conclusions are well-stated.

---

## Round 0.2 · accepted · Accept

Myself and one of the reviewers who originally reviewed your manuscript have reviewed your revised submission. We are both satisfied that you have addressed our, and the other reviewers', comments and concerns. I am therefore happy to accept your article for publication in PeerJ.

The only minor comment that I request you address at this point is a sentence you added with your revisions. On line 124 you state "We didn’t personally observe such effects however and believe that the treatments were largely independent.” I am not sure you can claim this – please delete this sentence. At the very least, please alter "didn't" to "did not".

Reviewer 2 ·

Basic reporting

ok

Experimental design

ok

Validity of the findings

ok

Additional comments

Overall, I like how authors have changed the manuscript in order to make their conclusions more prudent since they are based on a poor dataset. As they are now, figures better represent the data, also the map is definitely useful.